# Calcium imaging with genetically encoded indicators in behaving primates

Eyal Seidemann[1,2,3]*, Yuzhi Chen[1,2,3], Yoon Bai[1,2,3], Spencer C Chen[1,2,3], Preeti Mehta[3,4], Bridget L Kajs[3,4], Wilson S Geisler[1,2], Boris V Zemelman[3,4]

[1]Center for Perceptual Systems, University of Texas, Austin, United States; [2]Department of Psychology, University of Texas, Austin, United States; [3]Department of Neuroscience, University of Texas, Austin, United States; [4]Center for Learning and Memory, University of Texas, Austin, United States

**Abstract** Understanding the neural basis of behaviour requires studying brain activity in behaving subjects using complementary techniques that measure neural responses at multiple spatial scales, and developing computational tools for understanding the mapping between these measurements. Here we report the first results of widefield imaging of genetically encoded calcium indicator (GCaMP6f) signals from V1 of behaving macaques. This technique provides a robust readout of visual population responses at the columnar scale over multiple $mm^2$ and over several months. To determine the quantitative relation between the widefield GCaMP signals and the locally pooled spiking activity, we developed a computational model that sums the responses of V1 neurons characterized by prior single unit measurements. The measured tuning properties of the GCaMP signals to stimulus contrast, orientation and spatial position closely match the predictions of the model, suggesting that widefield GCaMP signals are linearly related to the summed local spiking activity.

*For correspondence: eyal@
austin.utexas.edu

Competing interests: The authors declare that no competing interests exist.

## Introduction

Understanding the contribution of a particular cortical area to mental phenomena such as perception necessitates analysis of neural activity across multiple spatial and temporal scales and over broad regions of cortical surface. For example, in primate primary visual cortex (V1), the smallest localized stimulus activates an area of multiple $mm^2$ (*Hubel and Wiesel, 1974*; *Palmer et al., 2012*; *Van Essen et al., 1984*), a region containing millions of neurons. Current techniques for measuring neural activity in behaving animals suffer from a fundamental tradeoff between resolution (spatial and temporal) and coverage, such that, even in a simple perceptual task with a localized stimulus, no single technique can measure the activity of all potentially relevant neurons in a given cortical area with single neuron resolution. Therefore, to advance our understanding of the neural basis of behavior, we need to (i) complement techniques that provide single-neuron access over a limited region with techniques that capture neural population responses over a large region, and (ii) develop computational tools that describe the mapping between neural signals measured using different techniques and combined across different spatial scales.

The macaque is an important animal model for human perception, cognition and motor planning, but many of the advanced optical and genetic tools available in rodents and simpler organisms are unavailable in the macaque. Beyond an early study with limited success (*Heider et al., 2010*), there are no reports of successful use of genetically encoded reporters in the macaque. Therefore, there is a growing need to translate advanced optical-genetic techniques to primates and other higher mammals.

**eLife digest** An important question in brain research is how neurons and the circuits they form process information to produce behavior. To understand what happens in a human brain, it is necessary to study a brain of similar complexity, such as that of a primate. Examining how the neurons in a brain region called the visual cortex process information about what we see is especially informative. This is because animals can be taught to perform different visual tasks, and because the visual cortex is relatively easy to access. In principle, therefore, it should be possible to use modern genetic and imaging techniques to study the primate visual system, but, until now, that has not been the case.

Like much of the brain, the visual cortex consists of different classes of neurons that can excite, inhibit or modulate the activity of neighboring neurons. One way to study how these different classes of neurons interact with each other is to alter the animal's DNA, such that only one cell type stands out during the experiment, allowing its role in the brain to be closely monitored. This technique has been used to study the interactions among neurons in the rodent brain, because rodent DNA is easy to alter. However, it is not easy to manipulate primate DNA.

Seidemann et al. have, therefore, developed a new technique that can target a specific class of neurons, allowing the activity of just these cells to be distinguished from the rest. The method uses specially designed harmless viruses to produce foreign proteins in the excitatory neurons of the visual cortex in an adult macaque. The optical properties of the proteins change when the neuron they are in is active, allowing the activity of the excitatory neurons to be detected and tracked in awake animals while they perform a visual task.

Previously, the activity of neurons in the primate visual cortex could only be measured using dyes that indiscriminately reported the activity of all the neurons present. Seidemann et al. found that, in addition to being more selective than the dye-based method, the new technique also more accurately depicted neuronal action potentials, which are the primary units of information in the brain.

Seidemann et al. now plan to use a similar method to study the activity of the inhibitory neurons of the primate visual cortex. Further examination of both excitatory and inhibitory neurons at much higher magnification, using a different microscopy technique, will also reveal more subtle features of their responses during visual tasks.

Here we report results of a new method using the genetically encoded calcium indicator GCaMP6f (*Chen et al., 2013*) for chronic imaging of neural population responses at the scale of orientation columns in V1 of behaving monkeys. Macaque V1 is arguably the most studied and best characterized cortical area in the primate brain. An important feature of macaque V1 is its topographic organization, which includes a large-scale retinotopic map (*Adams and Horton, 2003*; *Van Essen et al., 1984*; *Yang et al., 2007*) and a finer scale orientation map (*Bonhoeffer and Grinvald, 1991*; *Chen et al., 2012*; *Hubel and Wiesel, 1963*). The topographic organization of the orientation map at the columnar scale, which is absent from rodent V1 (*Ohki et al., 2005*), together with the existing extensive knowledge of the tuning properties of V1 neurons, make macaque V1 an ideal testbed for new imaging techniques and for studying the link between locally pooled neural population responses and single neuron responses.

## Results

### Quality and stability of the GCaMP signals

Our first goal was to find a viral vector system and an injection protocol that provide reliable and stable GCaMP expression for imaging in macaque V1. To do this, we took advantage of our expertise in long-term maintenance (>6 months) of a large (18 mm diameter) cortical window with direct physical and optical access to the awake macaque brain (*Figure 1*). With this window, it is possible to test in parallel more than a dozen combinations of vectors and injection parameters. Initial injections were performed with enhanced green fluorescent protein (EGFP) to test recombinant adeno-

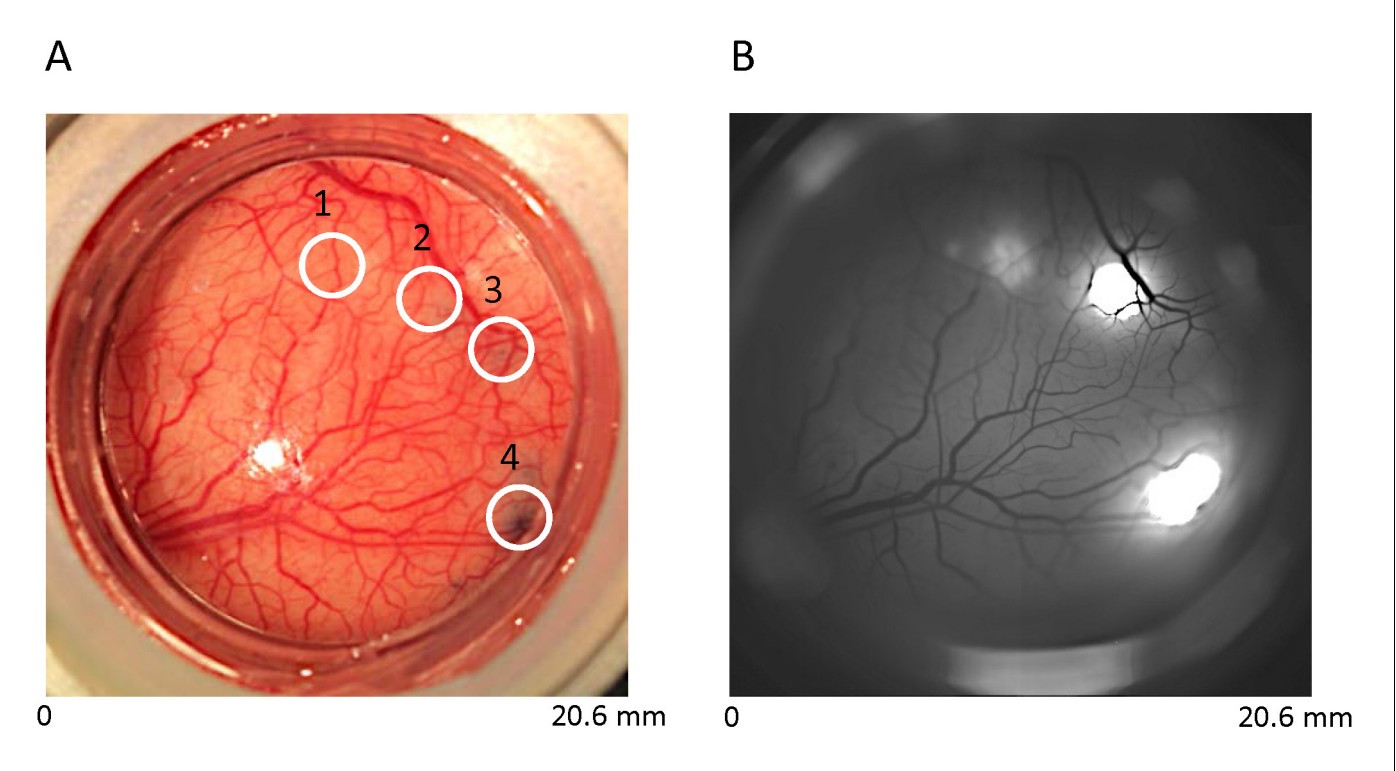

**Figure 1.** Construct-specific EGFP expression in V1. (**A**) Cortical vasculature with four injection sites in one cranial window over V1 of a macaque monkey: 1, rAAV9:mSYN-EGFP; 2, rAAV8:CaMKII-EGFP; 3, rAAV8:mSYN-EGFP; 4, rAAV1:CaMKII-EGFP. Blue stain at sites 2, 3 and 4 reflect the location of viral injections with trypan blue (see Experimental Procedures). Viral injection at site 1 was performed three months earlier, and hence shows minimal blue stain. (**B**) In vivo epifluorescence image three months post-injection at sites 2–4. Fluorescence at sites 1, 2 and 4 was prominent, but not at site 3. Fluorescence using the human synapsin promoter construct is not shown.

associated virus (rAAV) serotypes and promoters that have proven effective in the rodent (see Experimental Procedures). Following injection, each chamber was monitored weekly with epi-fluorescence imaging to track transgene expression levels. Injection sites were easily tracked based on high magnification images of the vasculature. Based on preliminary measurements, we selected serotype 1 rAAVs and CaMKIIα promoter (*Dittgen et al., 2004*) for expressing GCaMP. Glass micropipettes were used to inject small volumes (1–2 μL) of rAAV:CaMKII-GCaMP6f into V1 of three macaque monkeys. Six to seven weeks post-injection, high levels of transgene expression could be observed over an area of ~6–10 mm$^2$ per injection site (*Figure 2A*). Based on in situ hybridization analysis of gene expression at the injection site in one animal, approximately 80 percent of cells were CaMKIIα-positive and, therefore, excitatory. Further, all neurons expressing GCaMP were CaMKIIα-positive, attesting to the fidelity of the viral promoter (*Figure 3*). There was no evidence of apoptotic neurons, as judged by Hoechst 33342 staining 10 weeks post-injection (not shown). Once GCaMP fluorescence had been detected, we used widefield epi-fluorescence imaging to characterize the nature of the GCaMP signals carried by populations of V1 neurons while the monkeys performed a fixation task.

Our second goal was to assess the quality and stability of the widefield GCaMP signals. While the monkeys maintained gaze at the center of the screen, high contrast sinusoidal grating stimuli at 12 equally spaced orientations were flashed at 4 Hz for 4–6 cycles. Stimulus trials were mixed with blank fixation trials, and all conditions were randomly interleaved. Clear GCaMP signals were entrained on each stimulus cycle (*Figure 2B*). These signals provided a robust readout of visual responses with a high signal to noise ratio (SNR), which was stable for up to six months (*Figure 4A*).

To determine if the detected signals can be used to extract orientation maps, we measured the GCaMP responses across 10 repeats per orientation, computed as the 4 Hz Fourier amplitude of the average GCaMP signal acquired at each location, and applied a bandpass spatial filter (0.8–3 cycle/

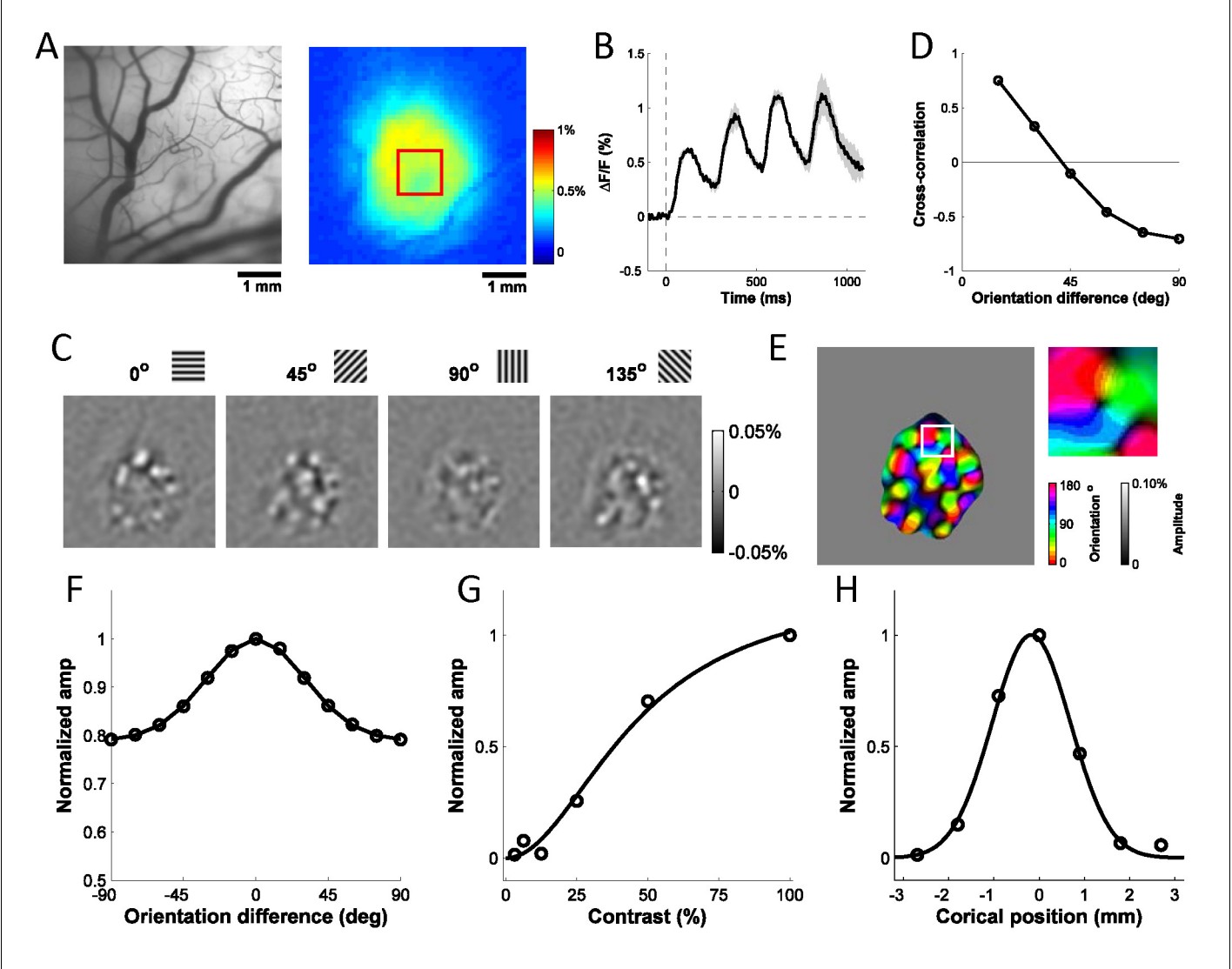

**Figure 2.** Columnar-scale GCaMP signals from V1 of a behaving macaque. (A) Vasculature (left) and GCaMP signal (right) at one injection site. (B) Average time course of GCaMP response to a flashed grating. Shaded area ± SEM. (C) Spatial pattern of orientation selective GCaMP signals obtained by bandpass filtration of the response maps to 4 orientations (out of 12 evenly spaced orientations). (D) Pairwise correlations between all 12 orientation maps as a function of stimulus orientation difference. (E) Orientation map with an insert showing a pinwheel. (F–H) Examples of GCaMP orientation tuning (F), contrast tuning (G) and position tuning on the cortex (negative towards fovea) (H).

mm) to remove non-orientation-selective responses and high frequency noise (*Figure 2C*). To verify that the observed maps reflected orientation columns, we computed the Pearson correlation between all pairs of maps, and then averaged the correlations across all pairs of maps with the same stimulus orientation difference (*Figure 2D*). As expected from the known structure of orientation maps in V1, the pairwise correlations changed smoothly from positive values for maps produced by nearby orientations to negative values for maps produced by orthogonal orientations. We used the maps produced by each orientation to compute the composite orientation map, in which color indicates the preferred orientation and saturation the strength of orientation tuning (*Figure 2E*). This composite map reveals the semi-periodic organization of the orientation map, including the orientation pinwheels (see inset) (*Bonhoeffer and Grinvald, 1991*). To determine the consistency of these composite orientation maps across days and between GCaMP and voltage sensitive dye (VSD), we converted the color map of preferred orientation to a grayscale map in the range of [−1, 1] by taking the sine of the preferred orientation times two (*Figure 4C*). The converted orientation maps are

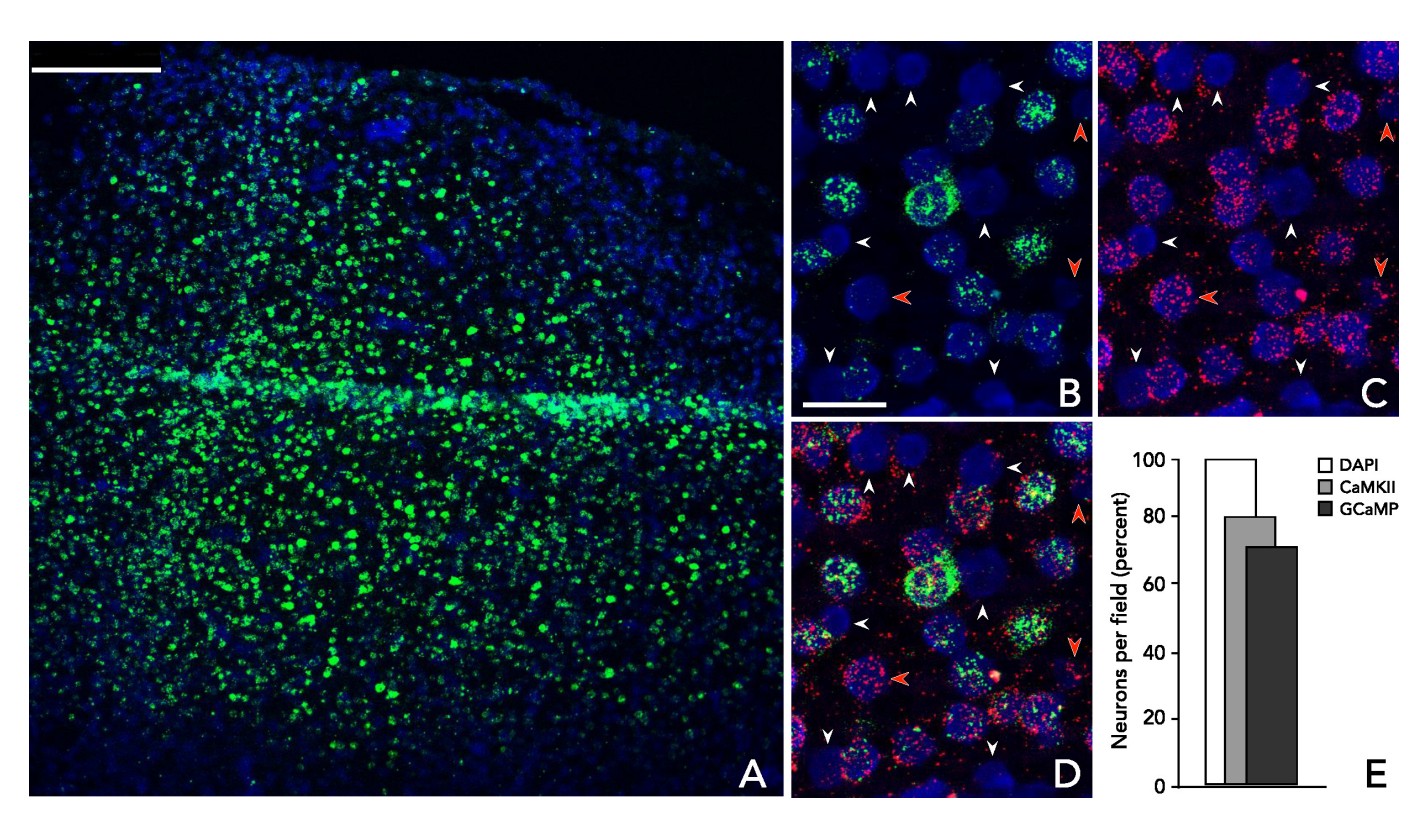

**Figure 3.** Analysis of virus-mediated GCaMP expression. Multiplexed in situ hybridization using probes to GFP (green, to detect GCaMP) and CaMKIIα (red) in tissue samples from a GCaMP6f expressing site in a third macaque 10 weeks after viral injection. Cells are identified using DAPI nuclear stain. (A) Low magnification image of a representative section within rAAV:CaMKII-GCaMP6f injection site. (B–C) Example GCaMP and CaMKIIα expression patterns, respectively. Red arrows: CaMKIIα(+)/GCaMP(-) cells; white arrows: CaMKIIα(-)/GCaMP(-) cells. (D) Overlayed hybridization patterns: 24 of 31 cells are CaMKIIα positive (77%), 20 of 24 CaMKIIα positive cells co-express GCaMP (83%). (E) Aggregate results. In a field of view containing 232 cells, 185 (80%) were CaMKIIα(+); of these, 168 (91%) were GCaMP(+) and 17 (9%) were GCaMP(-); none was CaMKIIα(-)/GCaMP(+). Scale bars: (A) 250 μm; (B–D) 25 μm.

consistent with maps obtained from the same region with VSD imaging and were stable for several months (*Figure 4B–E*).

## Tuning properties of widefield GCaMP and VSD signals to stimulus orientation, position and contrast

A major challenge with any novel technique is to understand the nature of the measured signals. Our next goal was to understand the relationship between widefield GCAMP responses and the responses of single neurons. This relationship is difficult to predict because the widefield measurements may reflect calcium signals due to spikes as well as calcium signals due to synaptic potentials in dendrites. Even if the GCAMP signals are dominated by spikes, it can be difficult, and sometimes impossible, to predict properties of population responses from single-unit measurements (e.g., *Seidemann et al., 2009*). For example, in the present case, it is possible that the spike-to-calcium nonlinearities in single neurons (e.g., *Akerboom et al., 2012*; *Chen et al., 2013*) average out when pooling responses over many heterogeneous neurons, leading to widefield GCaMP signals that are linearly related to the summed local spiking activity.

As a first step in characterizing the relationship between widefield GCAMP responses and single unit responses, we measured the tuning properties of GCaMP signals to three fundamental visual stimulus dimensions, and compared these tuning properties to those obtained with widefield VSD imaging. We then developed a computational model that sums the spiking (and subthreshold)

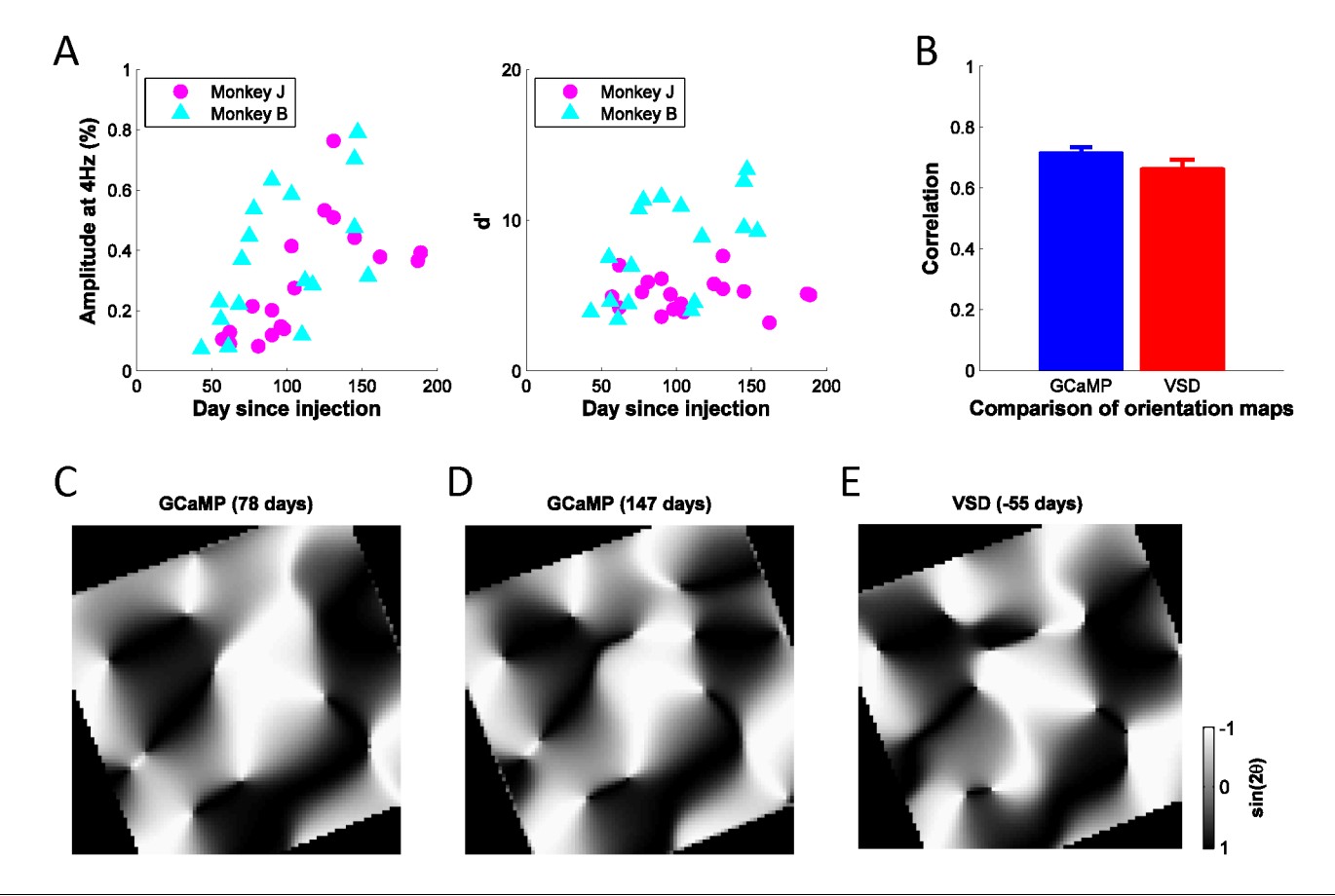

**Figure 4.** Stability of GCaMP signals over time and cortical map. (**A**) Signal amplitudes and d' (signal amplitude over signal standard deviation) to large flashed gratings over a period of ~200 days after viral injection for two monkeys. Fluctuations in signal amplitude and quality are mainly due to variations in the clarity of the aCSF fluid in the chamber. (**B**) Average correlation coefficients between any two GCaMP orientation maps (N = 8, blue bar), or between one GCaMP map (N = 8) and one VSD map (N = 2, red bar), where 'N' indicates separate experiments. Error bars ± SEM. Prior to computing correlations, maps were aligned based on the vasculature. (**C**), (**D**) Two examples of GCaMP orientation maps at the same site, but measured at different times after viral injection. The preferred orientation at each pixel is converted to the range of [−1, 1] by taking the sine of twice the preferred orientation. The correlation coefficient between the two gray-scale maps is 0.80. (**E**) One example of VSD orientation map at the same site as (**C**) and (**D**), but 55 days before viral injection. The correlation coefficients between the map in (**E**) and the maps in (**C**) and (**D**) are 0.71 and 0.78, respectively.

activity of a local population of V1 neurons, characterized by prior single unit measurements, and compared the model's predictions with the observed tuning properties of GCaMP and VSD signals.

We examined the tuning properties of V1 GCaMP responses across stimulus orientation, contrast and position. To compute orientation tuning, we measured how the response at each imaged cortical location changed as a function of the difference between stimulus orientation and the location's preferred orientation, and then averaged these curves across all locations in our region of interest (ROI; *Figure 2F*). The orientation tuning curve contains an orientation selective component and a large non-selective component, reflecting the fact that the local GCaMP signal pools responses from an area with a heterogeneous population of neurons (see below). Next, we examined how V1 responses in the same region change as a function of stimulus contrast (*Figure 2G*). The response was weak at low contrasts (<12%) and increased monotonically at higher contrasts. Finally, to examine position tuning, we measured GCaMP responses in a small ROI to localized Gabor stimuli (sine-wave grating in a Gaussian window) presented at different distances from the ROI's location in the visual field. Prior to these experiments, we used VSD imaging to obtain a precise retinotopic map of the entire recording chamber (*Figure 5*; *Yang et al., 2007*). We used the retinotopic map to

estimate, for each stimulus position, the cortical distance (in mm) between the center of the region activated by the stimulus and the center of our ROI. The GCaMP responses dropped rapidly as the stimulus' representation deviated from the recorded ROI's location, and no responses were observed when the distance exceeded 2 mm (*Figure 2H*). Similar GCaMP tuning curves for contrast, position and orientation were observed across multiple experiments spanning several months in two monkeys (*Figure 6A–C*, blue symbols; thin blue curves show results from two monkeys separately).

To better understand the nature and source of the widefield GCaMP signals in macaque V1, we compared the GCaMP tuning curves with those obtained using VSD imaging. Recent results suggest that local VSD signals in behaving monkeys reflect the membrane potential of cortical neurons pooled over a Gaussian shaped region with a space constant of ~200 µM (*Chen et al., 2012*). This pooling is likely to be primarily caused by lateral spread of neural processes—a small volume of cortical tissue contains axons and dendrites of neurons whose somata can be located several hundred microns away (*Chen et al., 2012*). Here we assume that this pooling area is comparable for widefield imaging of VSD and GCaMP signals.

What is the expected relation between widefield GCAMP and VSD signals if the GCaMP signal measures primarily the locally pooled spiking activity while the VSD signal measures locally pooled membrane potential? Due to the threshold nonlinearity between membrane potential and spiking activity in single neurons (*Priebe and Ferster, 2008*), stimuli that produce weak or moderate membrane potential response elicit almost no spiking activity. Therefore, the spiking responses of V1 neurons are more narrowly tuned for orientation and position and are less sensitive to low contrasts (*Priebe and Ferster, 2008*). In addition, the spike threshold nonlinearity causes spiking activity to have a smaller non-orientation-selective response relative to membrane potential (*Priebe and Ferster, 2008*). Therefore, if the GCaMP signal measures primarily the locally pooled spiking activity, then we would expect GCaMP responses to be less sensitive to contrast, more narrowly tuned to orientation and position, and to have a smaller non-orientation-selective response than VSD signals. A

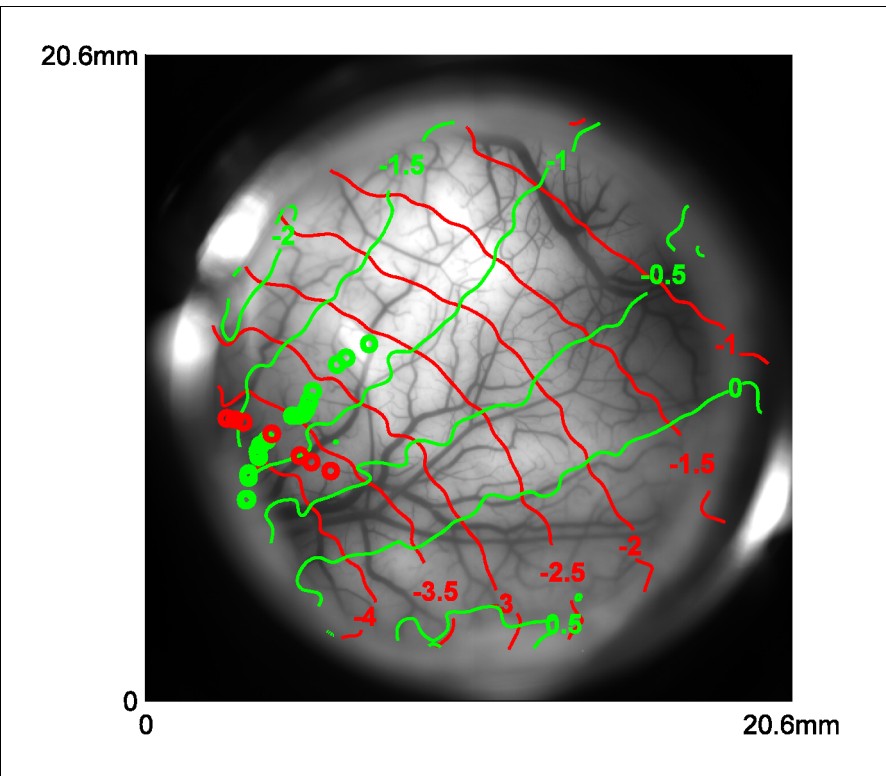

**Figure 5.** Retinotopic map obtained with VSD imaging and corresponding cortical locations of the centers of Gabor patches used in position tuning. Red and green curves represent the coordinates of X and Y in visual field in one monkey, respectively. Red and green circles represent the corresponding locations of the Gabors in vertical and horizontal trajectories, respectively.

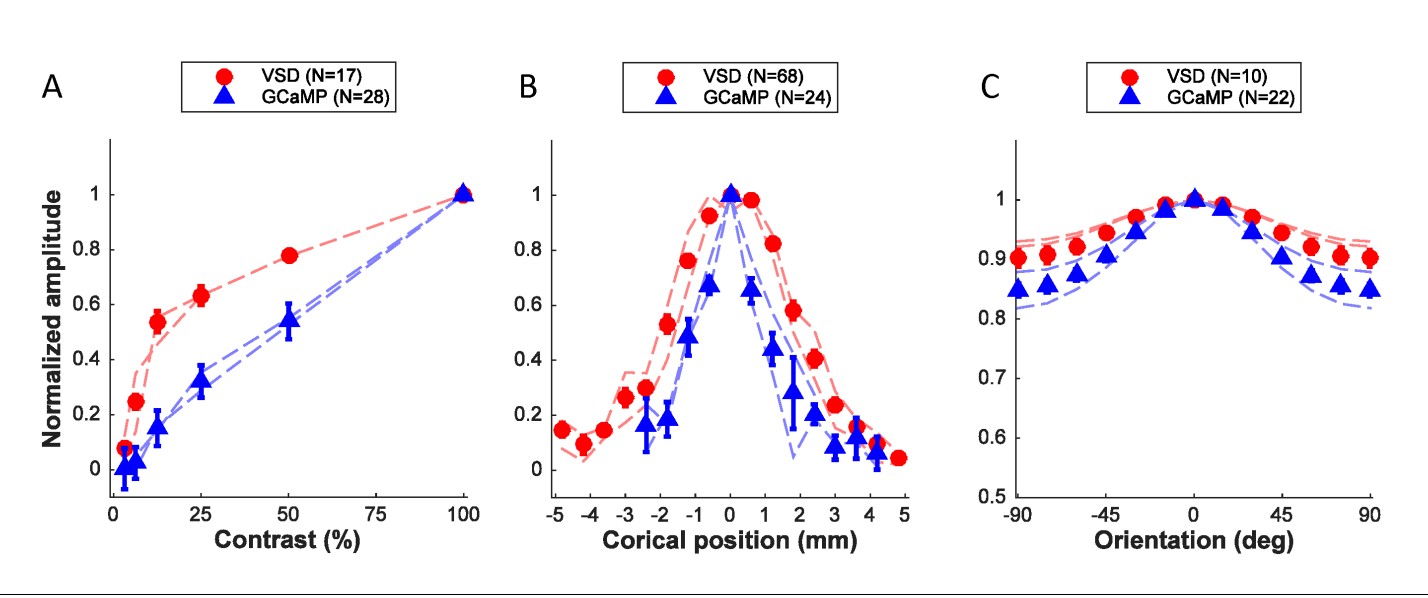

**Figure 6.** Quantitative comparison of neural population responses measured with GCaMP and VSD imaging. Summary of responses as a function of stimulus contrast (**A**), position (**B**), and orientation (**C**). Data points are normalized amplitudes averaged across multiple experiments. 'N' refers to experiments, with the exception of the VSD signals in panel B, where several position tuning curves were obtained simultaneously in the same experiment. In each panel results from the two animals that provided the majority of the data are indicated by the thin dashed curves separately for each animal (red – VSD; blue – GCaMP). Error bars ± SEM.

comparison of VSD (red symbols, *Figure 6A–C*; thin red curves show results from two monkeys separately) and GCaMP responses is largely consistent with these predictions, suggesting that the GCaMP signal may be proportional to the locally summed spiking activity.

## Linking widefield GCaMP and VSD signals and locally pooled neural activity

To further investigate the hypothesis that GCaMP responses measure pooled spiking activity while VSD responses measure pooled membrane potential, we evaluated a simple population model of GCaMP and VSD responses (*Figure 7*). This model capitalizes on the extensive measurements of single neuron activity that have been made in primate V1 over the last 40 years. Specifically, much is known about the distributions of the tuning properties of V1 neurons, as well as their functional organization (e.g., *Table 1*; *Hubel and Wiesel, 1974*; *Nauhaus et al., 2012*; *Van Essen et al., 1984*), making it possible for us to construct the model directly from single neuron measurements made by us and others. In the model, the membrane potential of neuron $i$ is controlled by its linear receptive field ($L_i$) and a contrast gain control mechanism, where gain decreases monotonically with contrast at a rate that is controlled by a semi-saturation parameter ($C_{50,i}$). Spikes are obtained by passing the membrane potential through a power-law spiking nonlinearity having an exponent $n_s$ (*Priebe and Ferster, 2008*). This is the standard descriptive model that has been used to summarize the response properties of single neurons in V1 (e.g., *Adelson and Bergen, 1985*; *Albrecht and Geisler, 1991*; *Heeger, 1992*).

To obtain the predicted GCaMP and VSD contrast response functions, we assumed (in agreement with the single unit literature) that the population of V1 neurons has a broad range of $C_{50}$ values. The relative contribution of neurons with a particular $C_{50}$ value to the pool is given by the distribution in *Figure 8A* (top), which approximates prior single unit measurements (*Table 1*). To obtain the predicted position tuning functions, we assume that individual neurons have a receptive field size (in mm) that is the average measured at this region of V1 (*Table 1*), and that the response at a given cortical location is the summed activity of a population of V1 neurons having scattered receptive field centers. The relative contribution to the sum of a neuron with a particular receptive field center is given by a spatial scatter function (*Figure 8B*, top). Similarly, to obtain the predicted orientation

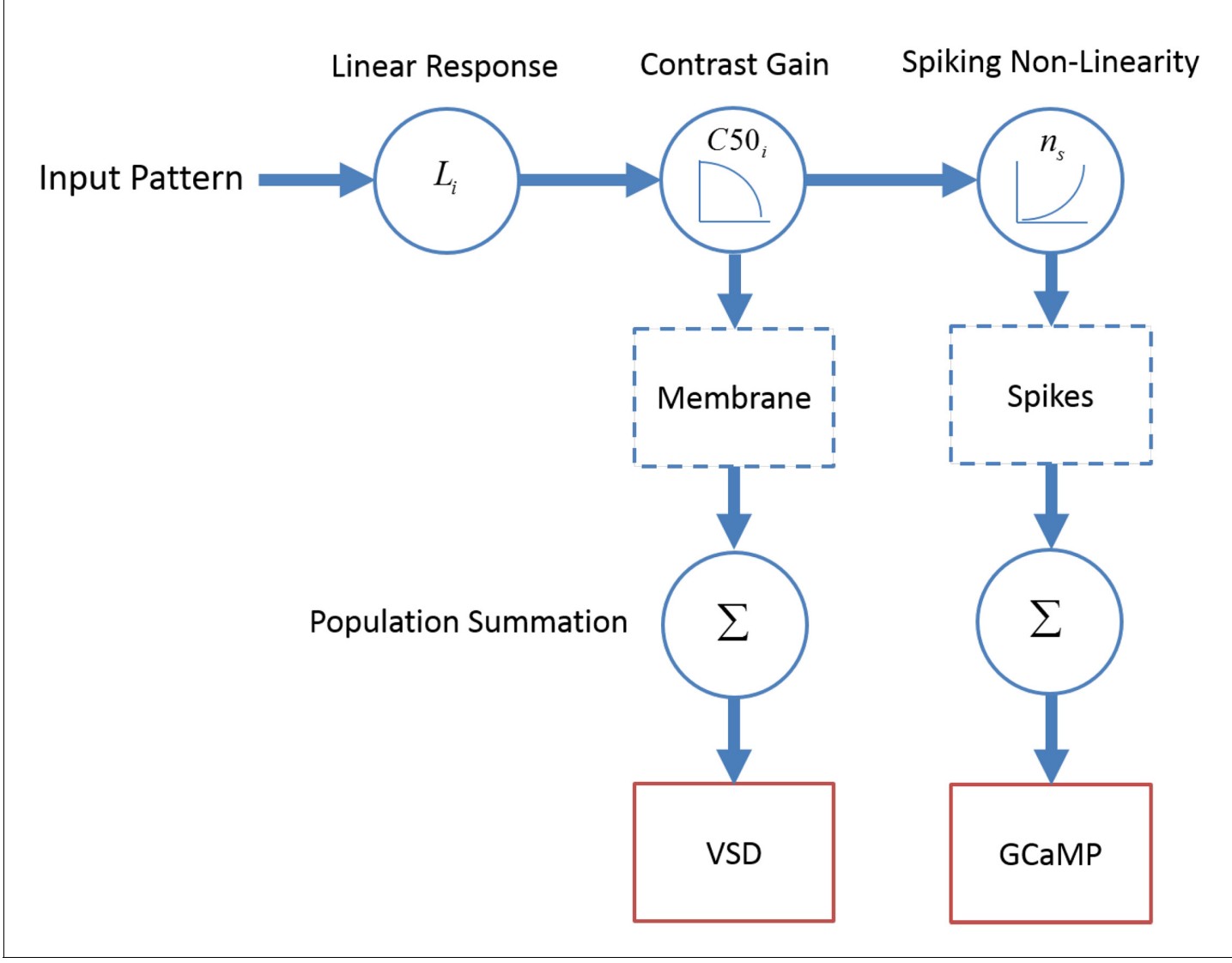

**Figure 7.** A computational framework relating GCaMP and VSD signals to single neuron membrane potential and spiking activity. Membrane potential of V1 neurons reflect a dot product of the visual input with the linear receptive field followed by contrast gain control. The neural mechanism producing spikes from membrane potential is described by a power-law non-linearity. As a first approximation, GCaMP and VSD signals reflect spiking activity and membrane potential, respectively, pooled over a local population of neurons with heterogeneous tuning properties.

tuning function we assume that individual neurons have the average orientation bandwidth of V1 neurons (*Table 1*) and that the response at a given location reflects the summed activity of a population of V1 neurons with different preferred orientations. The relative contribution to the sum of neurons with a particular preferred orientation is given by an orientation scatter function (*Figure 8C*, top). The position and orientation scatter functions reflect multiple factors that contribute to the broadening of the population tuning relative to that of single neurons (see Experimental Procedures). Due to the fine spatial scale of orientation maps (period of ~0.7 mm; (*Chen et al., 2012*; *Hubel and Wiesel, 1963*) and their semi-periodic nature, orientation scatter has a particularly large impact on the population orientation tuning curves. The widths of the position and orientation scatter functions are the only free parameters in the model. All other parameters, including the spiking nonlinearity exponent $n_s$, are taken from prior single unit studies (see Experimental Procedures and *Table 1*).

**Table 1.** Computational model parameters, values and literature references.

| Description | Symbol | Mean value | Refs. |
|---|---|---|---|
| Receptive field size full width half max | $W_x$ | 2.0 mm | *Hubel and Wiesel, 1974*; *Van Essen et al., 1984*; *Read and Cumming, 2003*; *Chen et al., 2012* |
| Orientation tuning bandwidth full width half max | $W_\theta$ | 40 deg | *De Valois etal., 1982*; *Vogels and Orban, 1990*; *Ringach et al., 2002*; *Nauhaus et al., 2008*; *Nowak and Barone, 2009*; *Palmer et al., 2012* |
| Fraction of tuned response | $f_\theta$ | 0.9 | *Ringach et al., 2002*; *Finn et al., 2007*; *Nowak and Barone, 2009*; *Palmer et al., 2012* |
| Spiking non-linearity exponent | $n_s$ | 3.0 | *Anderson et al., 2000*; *Hansel and van Vreeswijk, 2002*; *Miller and Troyer, 2002*; *Priebe et al., 2004*; *Tan et al., 2014* |
| Contrast semi-saturation distribution constant | $p_{50}$ | 0.5 <br> ($\overline{C_{50}}$ = 30%) | *Albrecht and Hamilton, 1982*; *Sclar et al., 1990*; *Geisler and Albrecht, 1997* |

The model's predicted tuning curves for contrast, position and orientation are qualitatively consistent with the observed GCaMP and VSD tuning curves (*Figure 8A–C*, bottom panels), providing support for the hypothesis that widefield GCaMP signals reflect the linearly summed local spiking activity, while widefield VSD signals reflect the linearly summed local membrane potential.

## Additional comparisons between GCaMP and VSD signals

To further compare the widefield GCaMP and VSD signals, we examined the dynamics of the visually evoked responses measured with these two techniques. *Figure 9A* shows the average time course of the GCaMP signal in the 22 experiments used to determine orientation selectivity of GCaMP signals (*Figure 6C*, blue symbols). The stimulus was presented for six cycles at 4 Hz (100 ms on, 150 ms off). Clear modulation to each stimulus presentation can be seen, but because the GCaMP signal does not return to baseline between presentations, the response builds up across the six stimulus presentations. *Figure 9B* shows the response collapsed across the six cycles. *Figure 9C–D* show the average time course of the VSD signal in the 10 experiments used to determine orientation selectivity of VSD signals (*Figure 6C*, red symbols). The stimulus was presented for five cycles at 5 Hz (60 ms on, 140 ms off). The VSD signal appears to rise and decay more rapidly, and does not show the same degree of buildup as the GCaMP signal. To compare the dynamics of the rising and falling edges of the GCaMP and VSD responses, we collapsed the responses across cycles, normalized the responses to their peak, and superimposed the GCaMP and VSD responses (*Figure 9E*). The latency of the VSD signal is shorter than the latency of the GCaMP signal. Similarly, the slope of the rising edge of the VSD signal is steeper than the slope of the rising edge of the GCaMP signal. Direct comparison of the slope and time-to-peak between these measurements is somewhat problematic because of the differences in the stimulus presentation protocol. To examine the dynamics of the falling edges, we aligned the two signals to the time of their peak (*Figure 9E*, right). The GCaMP signal falls off more slowly than the VSD signal, consistent with the differences in the buildup.

Finally, we compared the two signals in terms of their amplitude (*Figure 9F*, left) and signal-to-noise ratio (*Figure 9F*, right). The GCaMP signal is about five times larger than the VSD signal, but the two signals are comparable in terms of their sensitivity (d').

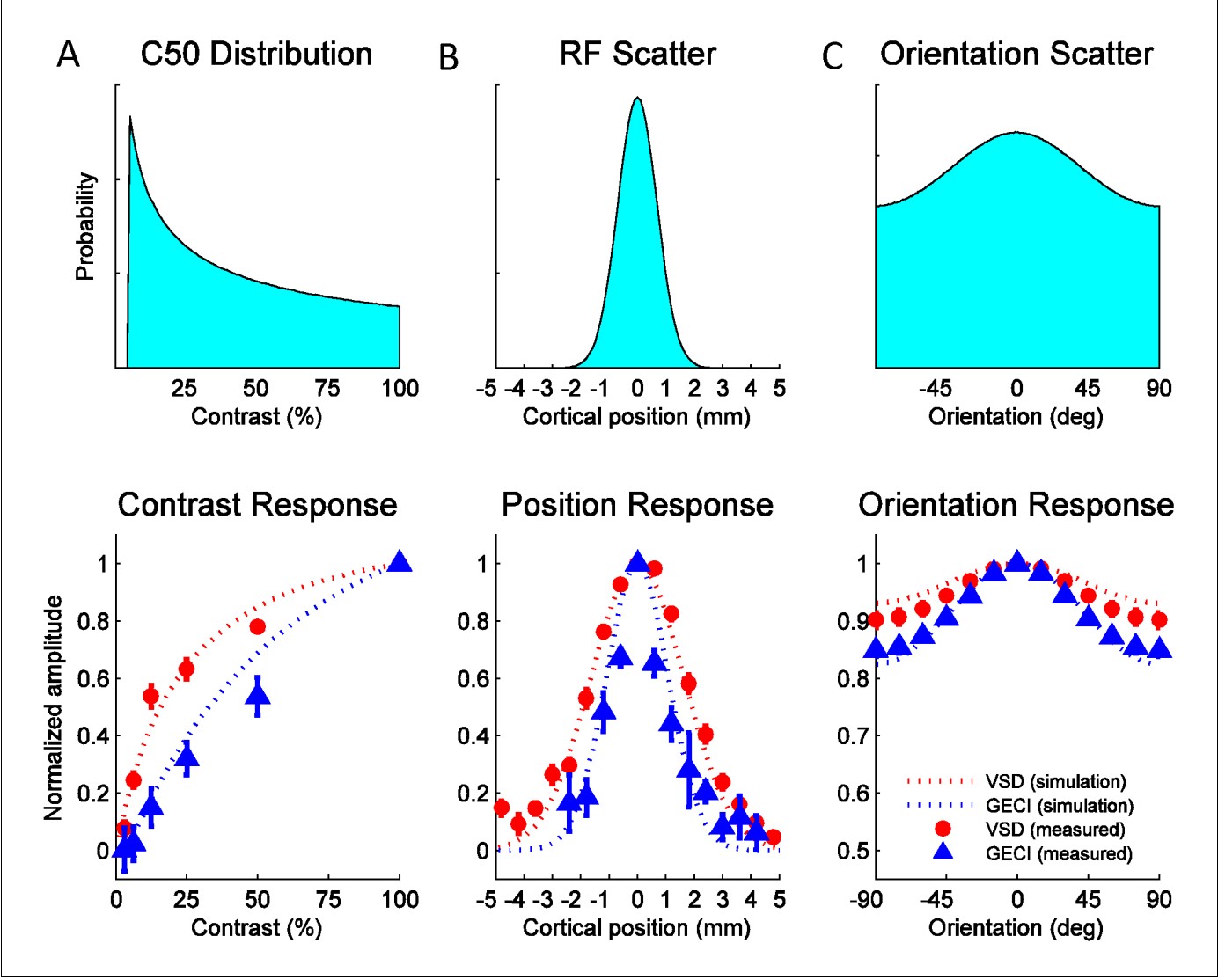

**Figure 8.** Widefield GCaMP signals in V1 of behaving macaques are consistent with the summed spiking activity. Simulated and observed tuning curves for stimulus contrast (**A**), position (**B**) and orientation (**C**). The computational framework in *Figure 7* was simulated with typical values for V1 receptive field (RF) parameters from the literature (see *Table 1*). Top panels show distributions of tuning properties while bottom panels show model predicted and observed tuning curves for GCaMP and VSD signals. Population pooling was made across neurons with different contrast semi-saturation (**A**), RF center position (**B**) and preferred orientation (**C**). In all stimulus dimensions (contrast, position and orientation), the simulation predicted GCaMP and VSD responses closely approximate the measured responses.

## Discussion

Here we report results of a new method for widefield imaging of GCaMP signals over multiple months from behaving macaque V1. This method provides high quality access to neural population activity at the scale of cortical columns. We measured the tuning properties of the GCaMP signals and compared them with those of VSD signals. We found that GCaMP signals are less sensitive to contrast and are more selective to position and orientation than VSD signals. We then developed a simple computational framework for modeling the relation between neural signals measured with different techniques and at different spatial scales. We used this framework to test the hypothesis that widefield GCaMP and VSD signals in V1 of behaving monkeys reflect the linearly summed local neural population activity at the level spiking activity and membrane potential, respectively. The general agreement between the model's predictions and our empirical results supports this hypothesis

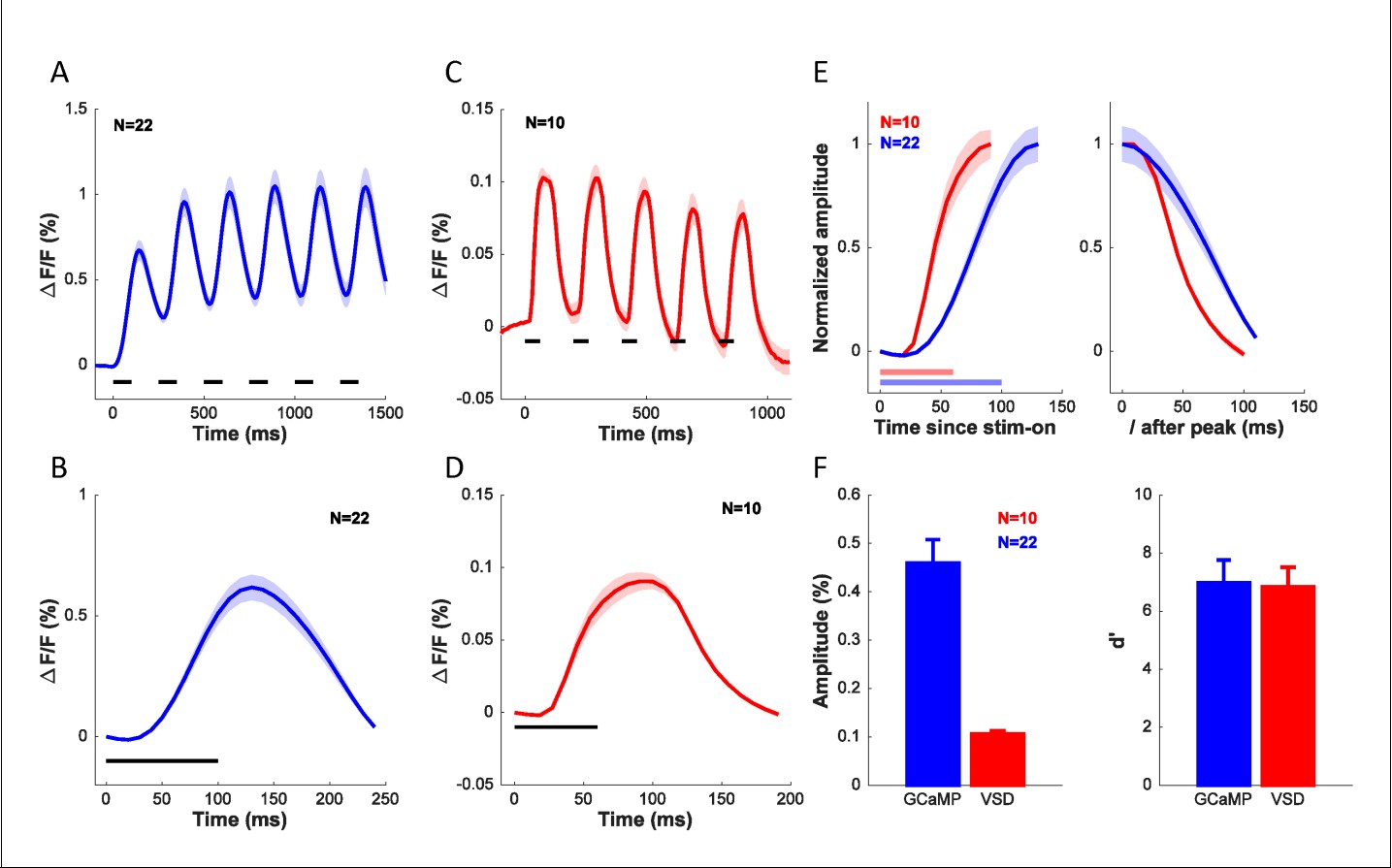

**Figure 9.** Comparison of dynamics, signal amplitude and signal sensitivity between GCaMP and VSD. (**A**) Time course of GCaMP signal in response to a grating flashed for 6 cycles at 4 Hz. (**B**) Average response from (**A**) collapsed across the 6 cycles. (**C**) Time course of VSD signal in response to a grating flashed for 5 cycles at 5 Hz. (**D**) Average response from (**C**) collapsed across the 5 cycles. (**E**) Dynamics of rising edge (left) and falling edge (right) of GCaMP and VSD. (**F**) Average amplitude (left) and sensitivity (right) of GCaMP and VSD signals. Horizontal lines in **A–E** indicates timing of stimulus presentation. When collapsing the response across cycles in **A–E**, the responses to each cycle were first anchored by subtracting the mean response in the first 36 ms (4 frames) for VSD and 50 ms (1 frame) for GCaMP after stimulus onset. Results from the same experiments as in *Figure 6C*.

and provides means to link neural signals measured at different spatial scales and with different techniques.

Because mammalian cortical representations are widely distributed, understanding the role of a particular cortical area in behavior necessitates complementing techniques that provide single-neuron access over a limited region with techniques that capture neural population responses over a large region of multiple mm$^2$. Chronic widefield imaging of genetically encoded indicators provides a promising approach to achieve this important goal. Our study represents several key advancements over a recent study using similar techniques in anesthetized ferrets (*Smith et al., 2015*). First, our experiments are performed in behaving primates that are an important and widely used model system. Second, we perform longitudinal measurements over several months, rather than acute experiments. Third, we develop a computational model that quantitatively links widefield measurements with single unit measurements and provides evidence that the widefield GCaMP signals are dominated by the summed spiking activity of local populations of neurons.

Widefield GCaMP imaging complements VSD imaging, another important tool for studying the dynamics of neural population responses in behaving primates (e.g., *Chen et al., 2006*; *Gilad et al., 2013*; *Michel et al., 2013*; *Muller et al., 2014*; *Seidemann et al., 2002*), because it measures a different aspect of neural population activity. GCaMP imaging offers several advantages over other widefield imaging techniques in behaving animals. The amplitude of the GCaMP signal is nearly an order of magnitude larger than that of VSD or intrinsic imaging (*Figure 9F*). The GCAMP signal has

much higher temporal resolution than intrinsic imaging (*Figure 2B*), although somewhat less than VSD imaging (*Figure 9*). In GCaMP imaging, a single viral injection procedure generates stable expression of a calcium indicator for many months (*Figure 4*). This eliminates the need for a repeated, challenging and variable-quality staining procedures necessary for VSD imaging. GCaMP signals are also less susceptible to bleaching than VSD signals, accommodating longer and more frequent imaging sessions. Thus, GCaMP imaging can be used to monitor the same cortical region over multiple months in behaving subjects, enabling new types of experiments, such as studies comparing neural and behavioral sensitivities in perceptual tasks with large ensembles of complex and naturalistic stimuli. Thus, chronic widefield GCaMP imaging in behaving animals is likely to become a widely used and powerful technique.

Our quantitative measurements of the tuning properties of the widefield GCaMP signals and our computational model suggest that these signals reflect the pooled spiking activity of layer 2/3 neurons rather than their pooled synaptic potentials or pre-synaptic inputs. Had the GCaMP widefield signal been dominated by synaptic potentials in dendrites or pre-synaptic inputs, their tuning properties should have matched those obtained with VSD. The large and consistent differences between the tuning properties of the widefield GCaMP and VSD signals (*Figure 6*), and the match between the model's predictions and the observed tuning properties (*Figure 8*), support the conclusion that the widefield GCaMP signal is tightly linked to the local spikes in layer 2/3. However, this result does not imply a lack of contribution from dendrites to the widefield GCaMP signal. Comparably tuned dendritic and somatic GCaMP6 signals have been recorded in rodent V1 using two-photon imaging (*Chen et al., 2013*), and we have obtained similar results in two-photon GCaMP6 experiments in macaque V1 (Nauhaus, Ko, Chen and Seidemann, unpublished observations). Therefore, stimulus-evoked dendritic GCaMP signals may be more correlated with the spiking output of layer 2/3 cells than with their synaptic inputs. Factors promoting dendritic GCaMP signals with tuning properties that are more similar to the spiking output than to the synaptic input could include backpropagating action potentials (*Larkum et al., 2007*; *Waters et al., 2003*), as well as nonlinear dendritic NMDA mechanisms (*Major et al., 2013*; *Xu et al., 2012*). Another important factor that is likely to contribute to the differences between the tuning properties of the widefield GCaMP and VSD signals is that the VSD signal is proportional to membrane surface area, while the GCaMP signal is proportional to cytosolic volume.

The current study represents one of the first steps in characterizing the relation between neural responses measured with different techniques and at different spatial scales. Our understanding of this relation needs to be improved in multiple ways. First, we need better quantitative measures of the relation between the optical signals, membrane potential and spiking activity at the level of single neurons. Second, we need a more detailed description of the heterogeneity and topographic organization of the tuning properties of single neurons in the cortex. Finally, we need better estimates of the various optical and biological factors that contribute to the spatial blur that underlies the pooled widefield optical signals.

Our successful use of GCaMP over multiple months opens the door for several exciting future technical advancements in behaving primates, including genetically targeted cell-type-specific imaging of neural populations and two-photon imaging of small neural populations with single-cell resolution. Indeed, preliminary results from two-photon imaging in anesthetized macaque expressing GCaMP6f show healthy neurons and characteristic paranuclear responses to visual stimuli, indicating that GCaMP imaging at cellular resolution can complement widefield imaging at the same anatomical location (Nauhaus, Ko, Chen and Seidemann, unpublished observations). Combining widefield and two-photon GCaMP imaging in the same animal over extended periods of time should help resolve many of the remaining questions regarding the relationship between neural responses measured over different spatial scales. Combining complementary empirical measurements with realistic computational models that interpolate fine-scale and large-scale measurements will provide unprecedented access to distributed neural responses in behaving subjects, leading to a deeper understanding of neural population coding in the brain.

## Materials and methods

### Injections and viral vectors

To screen viral vectors and optimize the injection protocol, we took advantage of our expertise in long-term maintenance of a large cortical window with direct physical and optical access to the awake macaque brain (*Figure 1*). Following injection, each chamber was monitored weekly with epi-fluorescence imaging to track transgene expression levels. Injection sites were easily tracked based on high magnification images of the vasculature. Initial test infections were performed using sero-type 1, 8, and 9 rAAVs encoding a mouse synapsin promoter and EGFP. Viruses were assembled using a helper-free system (Agilent Technologies, Santa Clara, CA), and purified on sequential cesium gradients according to published methods (*Grieger et al., 2006*). Viral titers were measured using a payload-independent quantitation technique (*Aurnhammer et al., 2012*). Typical titers were $>10^{10}$ genomes/microliter. To aid injection monitoring, 0.4% trypan blue (w/v) was diluted 1:10 in aCSF and added into the viral suspension. Injections were performed while the monkey was awake using methods similar to those recently developed for intracellular recording from behaving monkeys (*Tan et al., 2014*). A pulled glass needle (15 μm tip diameter) containing the suspension was lowered under surgical microscope guidance through an opening in the imaging chamber, punctur-ing the pia. Injection depths were measured from the point of contact between needle and brain surface. Virus was deposited at depths of 1.5, 1.0 and 0.5 mm (0.5 μl per site) using a positive dis-placement microinjector (Nanoject II). With the injector in slow mode, 0.5 μl of virus was delivered manually in 10 x 50 nl steps, with approximately 30 s pauses between steps. After virus was deliv-ered, the needle was left in place for 10 min before being withdrawn. The last injection normally pro-duced an easily visualized dye-stained region near the surface of the brain; the region continued to spread laterally in the minutes to hours following the injection, covering 5–10 mm$^2$ of cortex (*Figure 1A*). Importantly, the depth of the final injection was adjusted, such that staining was visible, but virus was not leaking from the injection site. Test experiments with rAAV:mSYN-EGFP produced regions of EGFP fluorescence (after 1–2 weeks) roughly matching the contours of dye patches (*Figure 1B*). Fluorescence signals associated with rAAV serotypes 1, 8 and 9 were qualitatively simi-lar. Hence, serotype 1 rAAVs was used for the GCaMP experiments.

We then tested several viral and mammalian promoters for the ability to express EGFP in primate cortex. Based on preliminary measurements, promoter strength matched that observed in the rodent (unpublished observations): mSYN ≈ hSYN<<CaMKIIα<CAG (*Borghuis et al., 2011*; *Dittgen et al., 2004*; *Niwa et al., 1991*; *Kügler et al., 2003*). Viral vectors rAAV:hSYN-GCaMP6f, rAAV:CamKII-GCaMP6f, rAAV:CAG-GCaMP6f were tested. The CAG vector produced high levels of unvarying basal fluorescence, consistent with compromised cell function. rAAV:CamKII-GCaMP6f at a titer of 1.3 x 10$^{11}$ viral genomes per microliter produced more robust GCaMP signals than the hSYN vector, and was, therefore, used for all experiments reported here. Future studies, including two-photon imaging, will more closely examine GCaMP signals obtained with hSYN, CamKIIα and other cell type-specific promoters. GCaMP fluorescence was observed 40 days following injection, peaking around 75 days, and remained constant for up to 190 days after injection (*Figure 4A*). The decrease in signal quality beyond this period was primarily caused by general deterioration of the health of the chamber. Future studies will examine the longer term stability of GCaMP expression.

### In situ analysis of GCaMP expression

Multiplexed hybridization to GCaMP and CamK2α transcripts was performed using the RNAscope system (Advanced Cell Diagnostics). Briefly, cortical tissue was collected using a 4 mm biopsy punch (Integra) and immediately frozen in OCT medium (Tissue Tek). Tissue blocks were cryosectioned at 12 μm (Leica CM3050S) and processed according to manufacturer instructions. Fixed and dehy-drated sections were co-hybridized with proprietary probes to GFP (to detect GCaMP) and CamKIIα, followed by selective fluorescence tagging. Signals in cells identified using DAPI staining were co-localized on a confocal microscope (Leica DM6000).

### Visual stimuli and behavioral task

Monkeys performed a visual fixation task. To obtain fluid reward, monkeys had to maintain fixation within a small window (<2 deg full width) centered on a small fixation point for 2–4 s. Eye position

was monitored by an infrared eye tracker (Dr. Bouis or EyeLink). To measure contrast and orientation tuning, we used a large sine wave grating (6 x 6 deg$^2$) centered over the receptive fields of the infected neurons. For position tuning we used a small Gabor patch ($\sigma$ = 1/6°; spatial frequency 2 cpd) presented along a linear trajectory at different distances in the visual field from the average receptive field of the neurons within a small ROI. For contrast and position tuning, visual stimuli were flashed 2–3 times per trial at 2 Hz (200 ms on; 300 ms off). Widefield measures of population responses with GCaMP and VSD contain a large non-orientation-selective component (*Figures 2F*, *6C*). Because the orientation selective signals are small, we had to increase our SNR in order to obtain high quality orientation maps and orientation-tuning curves. To increase the SNR, gratings were flashed at 4 Hz (100 on, 150 off) for GCaMP imaging and at 5 Hz (60 ms on, 140 ms off) for VSD imaging, in a period of 1 or 1.5 s. Additional tests verified that maps and tuning curves obtained with VSD at 4 and 5 Hz are identical (data not shown). Similarly, additional tests showed that the relation between the contrast and spatial tuning curves obtained with GCaMP and VSD imaging remains the same irrespective of the stimulus presentation frequency (data not shown). The mean luminance of the screen was maintained at 30 cd/m$^2$. All stimulus conditions were randomly interleaved and repeated at least 10 times.

## Imaging

All procedures have been approved by the University of Texas Institutional Animal Care and Use Committee and conform to NIH standards. The experimental techniques for optical imaging in behaving monkeys have been described in detail elsewhere (*Arieli et al., 2002*; *Chen et al., 2006*; *Shtoyerman et al., 2000*). Briefly, each animal was implanted with a metal head post and a metal recording chamber located over the dorsal portion of V1, a region representing the lower contralateral visual field at eccentricities of 2–5 deg (*Figure 5*). Craniotomy and durotomy were performed in order to obtain a chronic cranial window. A transparent artificial dura made of silicone was used to protect the brain while allowing optical access for imaging. We imaged GCaMP signals from V1 of two monkeys. VSD imaging was obtained from V1 of seven monkeys including one monkey that was subsequently used for GCaMP imaging (three monkeys contributed to position tuning, four monkeys for orientation tuning and two monkeys for contrast tuning) using methods described previously (*Chen et al., 2006*; *2008*). Epi-fluorescence imaging was performed with an imaging system (Imager 3001/M, Optical Imaging) using the following filter sets: GCaMP, excitation 470/24 nm, dichroic 505 nm, emission 515 nm cutoff glass filter; VSD, excitation 620/30 nm, dichroic 660 nm, emission 665 nm cutoff glass filter). Illumination was obtained with an LED light source (X-Cite120LED) for GCaMP or a QTH lamp (Zeiss) for VSD imaging. Data acquisition was time locked to the animal's heartbeat. Imaging was performed at 20 Hz for GCaMP imaging and 100 Hz for VSD imaging.

## Data analysis

Imaging signals were averaged across repeats, and the average time course in blank trials was subtracted from the response in each stimulus condition. For contrast and position tuning, neural signals were integrated over a 200 ms window starting 35 ms after stimulus onset. For orientation tuning, the response was the first harmonic amplitude obtained by FFT of the temporal response (as in (*Chen et al., 2012*). For contrast and orientation tuning of the GCaMP signal, tuning curves were computed over an ROI of 1.6 x 1.6 mm$^2$ centered at the location with the maximal d' (average signal amplitude over signal SD). For VSD, the ROI was the region with d'>3. For position tuning of the GCaMP and VSD signals, the tuning curves were computed over an ROI of 0.75 x 0.75 mm$^2$ centered at the retinotopic representation of the center stimulus. Each response curve was normalized by the peak value, then, averaged across experiments and monkeys (*Figure 6*).

## Model for relating GCaMP and VSD signals to neural activity

We examined how the measured GCaMP and VSD responses relate to neural activity using a simple model of V1 neural response (*Figure 7*). The model of membrane-potential and spiking activity is based on tuning parameter values of single V1 neurons measured by us and by others (see *Table 1*). We then simulated population pooling to obtain predictions that could be compared with the recorded GCaMP and VSD responses.

We modeled membrane-potential response for contrast ($L_c$), spatial position ($L_x$) and orientation ($L_\theta$) as follows:

$$L_c = \frac{c}{\sqrt{c^2 + C_{50}^2}} \qquad L_x = \exp\left(-2log4\left(\frac{x - x_0}{\hat{W}_x}\right)^2\right) \qquad L_\theta = \hat{f}_\theta \exp\left(-2log4\left(\frac{\theta - \theta_0}{\hat{W}_\theta}\right)^2\right) + \left(1 - \hat{f}_\theta\right)$$

where $c$ is the contrast level, $C_{50}$ is the contrast semi-saturation constant, $x$ is spatial position, $x_0$ is the center of the receptive field (RF), $\hat{W}_x$ is the full-width RF size at half maximum, $\theta$ is stimulus orientation, $\theta_0$ is the preferred orientation, $\hat{W}_\theta$ is the full-width orientation bandwidth at half maximum, and $\hat{f}_\theta$ is the fraction of tuned response.

Spiking response was the membrane-potential response raised to an exponent $n_s$:

$$R_c = L_c^{n_s} \qquad\qquad R_x = L_x^{n_s} \qquad\qquad R_\theta = L_\theta^{n_s}$$

As tuning parameters are most commonly reported for spiking activity, the equivalent values for membrane potentials (hatted parameters) were derived from spiking values by undoing the effect of the spiking non-linearity:

$$\hat{W}_x = W_x\sqrt{n_s} \qquad\qquad \hat{W}_\theta = W_\theta\sqrt{n_s} \qquad\qquad \hat{f}_\theta = 1 - (1 - f_\theta)^{1/n_s}$$

Next, we simulate population pooling in VSD and GCaMP imaging. The neurons contributing to the population responses at a given location are assumed to have a distribution of preferred position and orientation. The dispersion in preferred position and orientation results from multiple factors, including optical and biological blurring, biological scatter in tuning properties of neighboring neurons, and for position tuning, the size of the ROI.

The neurons contributing to the population response at a given location are assumed to have a distribution of contrast semi-saturation values. However, because there is no known topographic organization of semi-saturation values, the factors that cause the dispersion in preferred position and orientation have no effect on the distribution of semi-saturation values.

The distribution of contrast semi-saturation values ($w_c$) is modeled as power-function decay; the scatter in spatial position ($w_x$) is modeled with a normal distribution; and the scatter in orientation tuning ($w_x$) is modeled with a von Mises distribution:

$$w_c(C_{50}) = \frac{1}{K_c}\frac{1}{c_{50}^{p_{50}}} \qquad w_x(x_0) = \frac{1}{K_x}\exp\left(-\frac{1}{2}\left(\frac{x_0}{\sigma_{x_0}}\right)^2\right) \qquad w_\theta(\theta_0) = \frac{1}{K_\theta}\exp(\kappa_\theta\cos(\theta_0))$$

where $p_{50}$ controls semi-saturation contrast scatter, $\sigma_{x0}$ controls spatial scatter, $\kappa_\theta$ controls the orientation scatter, $K_c, K_x, K_\theta$ are constants normalizing the weights so that they sum to 1.0.

The pooled response is a weighted sum of individual neural responses. For membrane potential:

$$P_c = \sum_{C_{50}} w_c(C_{50})L_c(C_{50}) \qquad P_x = \sum_{x_0} w_x L_x(x_0) \qquad P_\theta = \sum_{\theta_0} w_\theta(\theta_0)L_\theta(\theta_0)$$

Similar equations for the pooled spiking response are obtained by replacing $L_c$, $L_x$ and $L_\theta$ with $R_c$, $R_x$ and $R_\theta$.

Single neuron parameters were set to typical values reported in the literature (see *Table 1*). At present, there is insufficient evidence in the literature to set the values for the spatial dispersion in preferred position and orientation. Thus, these two parameters, $\sigma_{x_0}$ and $\kappa_\theta$, were the only two allowed to vary in the model and were estimated by least squares (*Figure 8*).

## Acknowledgements

We thank Tihomir Cakic and Stefanie Esmond, as well as other members of Seidemann and Zemelman laboratories for their assistance with this project.

## Additional information

### Funding

| Funder | Grant reference number | Author |
|---|---|---|
| National Institutes of Health | EY016454 | Eyal Seidemann<br>Yuzhi Chen<br>Yoon Bai<br>Spencer C Chen |
| National Institutes of Health | EY261609 | Eyal Seidemann<br>Yuzhi Chen |
| National Institutes of Health | EY024662 | Wilson S Geisler<br>Eyal Seidemann |
| National Institute of Neurological Disorders and Stroke | NS094330 | Boris V Zemelman<br>Preeti Mehta<br>Bridget L Kajs |
| National Institute of Mental Health | MH100510 | Boris V Zemelman<br>Bridget L Kajs |
| Human Frontier Science Program | RGP0041 | Boris V Zemelman |
| National Eye Institute | EY026446 | Boris V Zemelman |
| National Eye Institute | EY026442 | Boris V Zemelman |

The funders had no role in study design, data collection and interpretation, or the decision to submit the work for publication.

### Author contributions

ES, BVZ, Conception and design, Acquisition of data, Analysis and interpretation of data, Drafting or revising the article; YC, Conception and design, Acquisition of data, Analysis and interpretation of data; YB, PM, BLK, Acquisition of data, Analysis and interpretation of data; SCC, Analysis and interpretation of data, Drafting or revising the article; WSG, Conception and design, Analysis and interpretation of data, Drafting or revising the article

### Author ORCIDs

Eyal Seidemann, http://orcid.org/0000-0003-2841-5948

### Ethics

Animal experimentation: All procedures were approved by the University of Texas Institutional Animal Care (protocol AUP-2013-00190) and Use Committee and conformed to National Institutes of Health standards.

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
