## [Decision Letter]

Thank you for submitting your article "Imaging columnar-scale neural population responses with genetically encoded calcium indicators in behaving primate V1" for consideration by *eLife*. Your article has been reviewed by two peer reviewers, and the evaluation has been overseen by a Reviewing Editor and David Van Essen as the Senior Editor. The following individual involved in review of your submission has agreed to reveal their identity: Aniruddha Das (Reviewer #2).

The reviewers have discussed the reviews with one another and the Reviewing Editor has drafted this decision to help you prepare a revised submission.

Summary:

Both reviewers agreed that the work represents an important technical advance. They find that this is a well written paper which demonstrates the first successful use of genetically encoded calcium indicator, GCAMP, to image neuronal activity on a mm scale in the awake monkey cortex. The authors directly compared widefield calcium imaging with imaging with Voltage Sensitive Dyes (VSD) in response to visual stimulation and used previously developed computational model to convincingly show that while VSD reflect subthreshold synaptic inputs, GCAMP measures spiking activity. Taking advantage of known properties and organization of neurons in area V1, the authors demonstrate that tuning of GCAMP signals to contrast, orientation and spatial position reflects locally pooled spiking activity.

Essential revisions:

The reviewers had a number of suggestions, focused largely on a few missing methodological details and clarifications. They also suggest adding a figure showing both GCAMP and VSD data for one of the animals as well as showing the comparison of the timecourses of the two types of signals and of their signal-to-noise ratios.

1) The data in Figure 6 appear to show grand averages for all population data. Please, add a figure comparing GCAMP and VSD data for one of the animals.

2) Provide a figure with timecourses and signal strength for the two types of measures.

3) Provide description of surgical methods, chamber implantation and behavioral paradigms, including fixation duration, fixation window size, eccentricity.

4) The scale bar in Figure 3 is confusing. The nominal value of 250 microns seems a bit low for panel A and too high for panels B-D, on the assumption that the pyramidal cell bodies in B-D are about 15-18 microns in size. Please double check the value, and provide a number for B-D.

5) Figure 4. Clarify the legend and explain whether the correlation was computed, between two maps in response to the same orientation or to any two maps. "N" is ambiguous: does N=8 refer to a single GCAMP data set?

6) Please clarify whether spread of dye = spread of virus.

---

## [Author Response]

1) The data in Figure 6 appear to show grand averages for all population data. Please, add a figure comparing GCAMP and VSD data for one of the animals.

To address this concern, we have added to Figure 6 curves for each individual animal (excluding animals with small number of experiments). We do not have enough data to compare the signals in the same animal, since the vast majority of the results were obtained from different animals. However, as indicated by Figure 6, the pattern of the results is consistent across animals.

2) Provide a figure with timecourses and signal strength for the two types of measures.

To address this concern, we added Figure 9, which compares the dynamics of the signals and their amplitude and SNRs.

3) Provide description of surgical methods, chamber implantation and behavioral paradigms, including fixation duration, fixation window size, eccentricity.

Additional technical details were added, as well as additional references describing some of the previously established methods.

4) The scale bar in Figure 3 is confusing. The nominal value of 250 microns seems a bit low for panel A and too high for panels B-D, on the assumption that the pyramidal cell bodies in B-D are about 15-18 microns in size. Please double check the value, and provide a number for B-D.

The value of the scale bar in panel A was verified and a proper scale bar was added to panels B-D. Figure legend has been edited, listing dimensions of both scale bars.

5) Figure 4. Clarify the legend and explain whether the correlation was computed, between two maps in response to the same orientation or to any two maps. "N" is ambiguous: does N=8 refer to a single GCAMP data set?

The correlations were computed between composite orientation maps after converting each map to the range of [-1, 1] by taking the sine of twice the preferred orientation. This is mentioned explicitly in the text and figure legend. “N” refers to number of experiments. In this case, all of the experiments were performed in the same animal in order to compare the GCaMP and VSD maps.

*6) Please clarify whether spread of dye = spread of virus.*

If the question refers to the spread of the VSD, the VSD is applied everywhere in the chamber and therefore stains all locations, while viral expression is restricted to an area with a radius of ~1-1.5 mm around the injection site. If the question refers to the spread of the trypan blue, then because the dye dissipates long before the virus-encoded proteins are expressed, there is no way to directly superimpose dye and GCaMP fields. However, since the dye is in the viral sample, and the dye field appears during the injection, it is reasonable to assume that the virus infects the brain volume marked by the dye. Panels in Figure 1, displaying dye pattern immediately after injection (A) and EGFP fluorescence three months later (B), support this conclusion. If this refers to the point-spread function, we added a sentence indicating that we assume that the point-spread functions of the VSD and GCaMP widefield signals are comparable. Our results appear to be consistent with this assumption.